# Liposomal Glutathione Augments Immune Defenses against Respiratory Syncytial Virus in Neonatal Mice Exposed in Utero to Ethanol

**DOI:** 10.3390/antiox13020137

**Published:** 2024-01-23

**Authors:** Theresa W. Gauthier, Xiao-Du Ping, Frank L. Harris, Lou Ann S. Brown

**Affiliations:** Department of Pediatrics, Division of Neonatal-Perinatal Medicine, Emory University, Atlanta, GA 30322, USA; xping@emory.edu (X.-D.P.); fharris@emory.edu (F.L.H.); lbrow03@emory.edu (L.A.S.B.)

**Keywords:** fetal ethanol exposure, alveolar macrophage, respiratory syncytial virus, oxidant stress, liposomal glutathione

## Abstract

We previously reported that maternal alcohol use increased the risk of sepsis in premature and term newborns. In the neonatal mouse, fetal ethanol (ETOH) exposure depleted the antioxidant glutathione (GSH), which promoted alveolar macrophage (AM) immunosuppression and respiratory syncytial virus (RSV) infections. In this study, we explored if oral liposomal GSH (LGSH) would attenuate oxidant stress and RSV infections in the ETOH-exposed mouse pups. C57BL/6 female mice were pair-fed a liquid diet with 25% of calories from ethanol or maltose–dextrin. Postnatal day 10 pups were randomized to intranasal saline, LGSH, and RSV. After 48 h, we assessed oxidant stress, AM immunosuppression, pulmonary RSV burden, and acute lung injury. Fetal ETOH exposure increased oxidant stress threefold, lung RSV burden twofold and acute lung injury threefold. AMs were immunosuppressed with decreased RSV clearance. However, LGSH treatments of the ETOH group normalized oxidant stress, AM immune phenotype, the RSV burden, and acute lung injury. These studies suggest that the oxidant stress caused by fetal ETOH exposure impaired AM clearance of infectious agents, thereby increasing the viral infection and acute lung injury. LGSH treatments reversed the oxidative stress and restored AM immune functions, which decreased the RSV infection and subsequent acute lung injury.

## 1. Introduction

In clinical studies, we previously reported that maternal alcohol use during pregnancy was associated with an increased risk of sepsis in both premature and term newborns [1,2]. Furthermore, in premature newborns weighing ≤ 1500 g at birth, maternal admittance of alcohol ingestion during pregnancy occurred in one third of pregnancies. In utero alcohol exposure in these low-birthweight premature newborns was associated with significantly increased odds of developing neonatal late-onset sepsis and bronchopulmonary dysplasia [1,2]. In addition to fetal alcohol spectrum disorder, other studies have demonstrated that alcohol consumption during pregnancy is associated with a range of adverse outcomes for the newborn, including spontaneous abortions [3], stillbirths [4,5], preterm birth [6,7], and low birthweight [8,9]. Despite these outcomes, our understanding of in utero alcohol’s detrimental effects on the developing lung remain limited. Studies to further understand the mechanisms underlying the risks of in utero alcohol exposure for the premature newborn, specifically in pulmonary health, are needed.

Using experimental animal models, we showed that ethanol (ETOH) exposure results in chronic oxidant stress in both the neonatal and the adult lung, which resulted in impaired cellular immune defenses of the resident alveolar macrophage (AM) against infectious agents. It is the decreased availability of glutathione (GSH), the primary antioxidant in the airspace, that is one key to the ETOH-induced immune dysfunctions of both newborn and adult AM [10,11]. Over time, the ETOH-induced decreases in GSH and subsequent chronic oxidant stress promote the release of immune suppressors like transforming growth factor β1 (TGFβ1) that subsequently result in compromised immune cells that are critical in determining disease outcomes [12]. These studies also demonstrated that strategies to improve GSH availability subsequently improved the immune responses of newborn and adult AM, including both viral and bacterial clearance.

For infants, respiratory syncytial virus (RSV) is the most common etiologic agent for acute respiratory infections, infections in the lower respiratory tract such as bronchiolitis and pneumonia, and the leading cause of hospitalization for infants less than 2 years of age [13,14,15,16]. Newborns at the greatest risk for severe infections include the premature newborn and infants with chronic respiratory diseases such as bronchopulmonary dysplasia related to prematurity [17]. When compared to adults, the newborn’s antioxidant status is low, but the generation of reactive oxygen species and lipid peroxidation products that occurs during a viral infection exacerbates their limited antioxidant availability, results in chronic oxidant stress, and contributes to the pathogenesis [18]. In premature newborns, their immature immune responses [19] also contribute to their risk of increased incidence and severity of viral infections such as RSV [20,21]. Indeed, lower respiratory tract infections with viruses such as RSV result in extended hospital stays, readmission to the pediatric intensive care unit, an increased need for oxygen and mechanical ventilation, and increased mortality [22]. Although airway epithelial cells are the primary site for RSV infection, innate immune responses by the mononuclear phagocytic system, including phagocytosis and the release of cytokines, chemokines, and other immune mediators, are critical for the resolution of pulmonary viral diseases [19,23]. As the resident mononuclear phagocytic cell in the lung and airspace, the AM is the key modulator of the local pulmonary immune response that is essential for RSV clearance [23,24].

The key role of antioxidant availability as a modulator of the pathogenic process was demonstrated in mouse models where antioxidant treatments attenuated RSV-induced oxidant stress as well as the viral burden [25,26]. In adult animal models, chronic alcohol exposure impairs multiple arms of immune defense mechanisms against respiratory viruses such as RSV [27,28]. In fetal lambs, in utero ETOH exposure predisposes the developing lung to RSV by altering host defenses via deranged surfactant proteins [29,30]. We recently demonstrated in our established fetal ETOH mouse model that in utero ETOH exposure impaired AM cellular capacity to defend against experimental RSV from the lung [31]. This impairment may potentially be modulated through the upregulation of the immunosuppressant TGFβ1, since we also observed that TGFβ1 directly impairs the capacity of AM to clear RSV [12].

Supplementation with an oral liposomal formulation of GSH (LGSH) has been shown to improve intracellular delivery of GSH and decrease oxidant stress in HIV subjects and patients with type 2 diabetes [32,33,34]. In addition, these studies demonstrated that LGSH treatments improved the immune responses of peripheral blood monocytes cells when treated in vitro with *Mycobacterium tuberculosis*, including improved bacterial clearance [33,34]. In ventilated preterm infants, a single intratracheal dose of LGSH increased pulmonary GSH pools and decreased oxidative stress [35]. The possibility that in vivo LGSH administration to the neonate exposed to ETOH in utero may improve immune defenses against viruses such as RSV is attractive, but has not been investigated. In the current study, we hypothesized that strategies such as LGSH treatment would augment the antioxidant GSH in the neonatal lung exposed to ETOH in utero, decrease AM oxidant stress, attenuate cellular immunosuppression, and improve neonatal AM immune defenses against an experimental RSV infection. The goals of the current study were to use an established mouse model of in utero ETOH exposure to (1) determine if a clinically relevant intervention, such as enteral LGSH, could protect against an experimental pulmonary RSV infection and (2) explore the potential mechanisms by which LGSH improved AM innate immune defenses against RSV.

## 2. Materials and Methods

Mouse model of fetal ethanol (ETOH) exposure. Our established mouse model of fetal ETOH exposure uses a continuous exposure of ETOH during pregnancy through a maternal Lieber-DeCarli liquid diet containing ETOH (BioServ, Frenchtown, NJ, USA) [31]. Female C57BL/6 mice were shipped from the vendor (Charles River, Burlington, MA, USA) and acclimated in the Emory Pediatrics animal facilities for one week. After breeding, the experimental liquid diet was started one day after visualization of the vaginal mucus plug. Pregnant dams were randomized to receive an isocaloric liquid diet ±25% ETOH-derived calories. For the ETOH group, the ETOH content of the diet was incrementally ramped up over a 1-week timeframe from 0% to 12.5% and then 25% of ETOH-derived calories. Food consumption was recorded daily and the liquid food was changed daily. The control group was pair-fed to the ETOH group with 25% of the calories obtained from maltose–dextrin. The only access to food during the experimental period was the assigned experimental liquid diet. The diet was continued throughout pregnancy and after spontaneous term delivery. Pups were kept with their respective dams and allowed to nurse *ad libitum*. All animals were used with protocols reviewed and approved by the Emory University Institutional Animal Care Committee (PROTO201800128) in accordance with NIH guidelines (Guide for the Care and Use of Laboratory Animals).

Respiratory syncytial virus (RSV) and RSV Plaque Assay. RSV clone rA2|19F was a generous gift from Martin L. Moore, PhD [36]. As we previously reported [12,31], RSV was propagated in Hep-2 (CCL-23) cells in minimum essential medium supplemented with 10% fetal bovine serum, penicillin (100 U/mL) and streptomycin (100 µg/mL) (Sigma, St. Louis, MO, USA). RSV was then harvested 6–7 days after the Hep-2 cell inoculation and sonicated on ice and centrifuged (500× *g*, 10 min at 4 °C). Plaque assays were performed to determine RSV titers (plaque forming units (PFUs)) by serially diluting the supernatant, infecting 24-well plates of Hep-2 cells (6 days at 37 °C and 5% CO_2_) and visualizing immunostaining.

Liposomal glutathione (LGSH) treatments and experimental Respiratory syncytial virus (RSV) in the neonatal mouse. To investigate a potential therapeutic role for LGSH during an in vivo RSV infection, we incorporated LGSH into our established in vivo mouse model of inhaled RSV [31]. LGSH (ReadiSorb Liposomal Glutathione) was a generous gift from Dr. Frederick T. Guilford (Your Energy Systems, Palo Alto, CA, USA). This preparation of LGSH contained reduced GSH (422.7 g/5 mL) plus purified water, glycerin, lecithin, and potassium sorbate. On day of life 10 (P10), mice pups (±in utero ETOH exposure) were treated with an oral gavage containing either LGSH (20 μL, 1.7 mg of L-glutathione) or saline (20 μL). Mouse pups were then given intranasal injections of RSV (Nanoliter Injector; 20 µL; each nasal nare; 2 × 10^5^ PFU) before they were returned to their respective dams. After 24 h, pups received an additional dose of ±LGSH (or saline) by oral gavage. All pups were then euthanized for analyses after 48 h.

Alveolar macrophage (AM) isolation. After euthanasia with intraperitoneal sodium pentobarbital, the pup trachea was identified under a dissecting microscope and cannulated with a 27 G catheter. The lungs were then serially lavaged via the catheter with 40 µL sterile saline (5 times) to remove the bronchoalveolar fluid lining the airspace. The initial lavage from each pup in a litter was pooled and centrifuged (402× *g*; 8 min) and the cell-free supernatant lavage fluid was saved for further analysis (noted below). The subsequent bronchoalveolar lavages (BALs) from each pup within the same litter were also pooled and similarly centrifuged. The cell pellets obtained from the initial and the subsequent lavages were resuspended in media (RPMI 1640 1 time) containing 10% fetal bovine serum and 1% antibiotics before they were pooled. The pooled cells from each litter represents an *n* of 1. Cell viability and cell count were determined with trypan blue stain (0.4%; Life Technologies, Grand Island, NY, USA). Pup AMs were cultured on slides, fixed with 3.7% paraformaldehyde and permeabilized with ice-cold methanol.

Plasma collection for systemic biomarkers of oxidant stress. After euthanasia with intraperitoneal sodium pentobarbital, blood samples were obtained from all pups via cardiac puncture and the samples pooled per experimental group and litter. Samples were spun and stored at −80 °C until batch analyses. Total plasma antioxidant capacity (AOC) was measured via colorimetric assay (MAK187, Sigma-Aldrich, St. Louis, MO, USA) and 8-hydroxyguanosine (8-OHdG) was measured by ELISA (DNA Damage Competitive ELISA kit, Life Technologies Corporation, Carlsbad, CA, USA).

Determination of the Respiratory syncytial virus (RSV) burden in the bronchoalveolar lavage fluid (BAL) and whole lung. All pups were euthanized for analyses 48 h after intranasal delivery of RSV. The lungs were then serially lavaged via the catheter with 40 µL sterile saline (5 times) to remove the fluid and cells from the airspace. To determine the RSV burden in the airways and alveolar space, the multiple BALs from pups within the same litter were pooled. The pooled BAL was then serially diluted in phosphate-buffered saline and plated for determination of RSV growth via the Hep-2 cell plaque assay, as we have previously described [31]. After the pup was euthanized, the right upper lobe of the lung was flash-frozen in liquid nitrogen and stored at −80 °C until batch analysis. The lobe was weighed, sterile phosphate-buffered saline (10 times the tissue weight) was added, and then the lobe homogenized on ice. After the samples were centrifuged (2000× *g*; 10 min; 4 °C), the supernatants were serially diluted in phosphate-buffered saline and then 100 μL of the homogenate dilution was similarly plated for determination of RSV growth. For the whole lung, RSV is presented as percentage of control of pfu/g lung tissue. The remaining isolated neonatal lung lobes were also flash frozen in liquid nitrogen and stored at −80 °C until batch analyses (see below).

Cellular immunostaining. The freshly isolated AMs were plated and fixed with 3.7% paraformaldehyde before permeabilization with ice-cold methanol. For assessment of RSV phagocytosis by the AM in vivo, we evaluated whole-cell RSV content via fluorescent immunostaining (1 h; a 1:100 dilution; Santa Cruz Biotechnology, Inc., Santa Cruz, CA, USA). Given the critical role of the antioxidant GSH in the AM immune phenotype, we evaluated whole-cell GSH via fluorescent immunostaining (1 h; 1:100 dilution; Abcam, Inc^®^, Boston, MA, USA). Since fetal ETOH exposure increases AM expression of the immunosuppressant TGFβ1 [11,31] and arginase-1 (Arg-1) [11], we also used immunostaining to evaluate whole-cell TGFβ1 and Arg-1 in the neonatal AM. Immunostaining for the neutrophil cell surface marker (Gr-1) was used to differentiate AM from polymorphonuclear leukocytes (PMNs). Cells were incubated with the respective primary antibody in a 1:100 dilution (Santa Cruz Biotechnology, Inc., Santa Cruz, CA, USA) for 1 h. After the slides were washed three times with phosphate-buffered saline over 5 min, and the secondary antibody (anti-goat IgG, ThermoFisher, Waltham, MA, USA) was added in a 1:200 dilution and further incubated for 45 min. Cellular fluorescence was quantified using fluorescence microscopy via ImagePro Plus for Windows, Version 4.5 and presented as mean relative fluorescence units per cell (RFUS/cell) ± S.E.M. as tallied from at least 25 cells/litter. To correct for autofluorescence, the background fluorescence of unstained AMs was subtracted from the RFUs obtained for each analysis.

Whole-lung analyses. In addition to determining the tissue RSV, the flash-frozen neonatal lung tissue was evaluated for the inflammatory PMN marker myeloperoxidase (MPO, ng/mL) with a commercially available ELISA (product # MBS2702122, MyBioSource, Inc., San Diego, CA, USA). In parallel experiments, the neonatal right upper lobe lung samples were weighed (designated wet weight) and then reweighed after desiccation by overnight incubation at 70 °C (designated dry weight) and the lung wet/dry weight ratio was determined. MPO concentrations (ng/mL) were normalized to lung sample wet/dry weight.

Statistical analysis. SigmaPlot software (Systat Software 14.5; San Jose, CA, USA) was used for statistical analysis and graph generation. ANOVA or Kruskal–Wallis ANOVA on ranks was used when appropriate to detect overall differences between groups. Student–Newman–Keuls or Dunn’s post hoc analysis was conducted for group comparisons as indicated. A value of *p* ≤ 0.05 was deemed statistically significant. Data are presented as mean ± S.E.M, where each *n* represents one litter.

## 3. Results

Fetal ethanol (ETOH) exposure decreased alveolar macrophage (AM) viral clearance and exacerbated the experimental lung Respiratory syncytial virus (RSV) infection in the exposed pup. In agreement with our previous study [31], in utero ETOH exposure impaired AM phagocytosis of inhaled RSV by 40% in the neonatal mouse pup when compared to the control pup (Figure 1). With the ETOH-induced decrease in AM in virus phagocytosis, there was an accompanying increase in the burden of RSV growth in the pup lung. Both the BAL (Figure 2A) and neonatal whole lung (Figure 2B) demonstrated a ~twofold increase in RSV growth in the ETOH pups when compared to the control pups.

Increased acute lung injury after Respiratory syncytial virus (RSV) inhalation in pups with in utero ethanol (ETOH) exposure. We next determined if the greater lung RSV burden associated with fetal ETOH exposure resulted in greater lung injury. With the lung wet/dry weight ratio as an indirect marker of lung edema, there was ~threefold increase in the ETOH + RSV group when compared to the control + RSV group (Figure 3A). Since pulmonary neutrophil infiltration can be associated with acute lung injury, the MPO content of the lung was used as a granulocyte marker. There was a ~fourfold increase in the MPO content of the lung in the ETOH + RSV group compared to the control + RSV group (Figure 3B). Using Gr-1+ as a marker of PMNs, we determined the percentage of cells that were neutrophils in the first lavage. In the ETOH + RSV group, there was ~twofold increase in neutrophils (GR-1+) in the alveolar space when compared to the control + RSV group (Figure 4). Taken together, these results suggest that compared to the control pup, fetal ETOH exposure resulted in a significant increase in these three indirect markers of acute lung injury after RSV inhalation.

Oxidant stress was exacerbated when the fetal ethanol (ETOH)-exposed pup was challenged with Respiratory syncytial virus (RSV). In the ETOH + RSV group, the plasma antioxidant capacity (AOC) was decreased ~35% when compared to the control + RSV group (Figure 5A). With plasma 8OH-dG as a marker of DNA oxidant damage, there was ~threefold increase in the ETOH + RSV group when compared to the control + RSV group (Figure 5B). When compared to the control + RSV group, fetal ETOH exposure superimposed on inhaled RSV also exacerbated AM oxidant stress, as shown by ~50% decrease in the cellular antioxidant GSH pool (Figure 6).

Increased markers of alveolar macrophage (AM) immune suppression with fetal ethanol (ETOH) exposure superimposed on experimental Respiratory syncytial virus (RSV). In the ETOH + RSV group, there was ~twofold increase in AM expression of the immunosuppressant TGFβ1 when compared to the control + RSV group (Figure 7A). Likewise, AM expression of Arg-1, an immunosuppressant, was increased ~twofold in the ETOH + RSV group when compared to the control + RSV group (Figure 7B).

In vivo liposomal gluthaione (LGSH) treatments improved alveolar macrophage (AM) function and improved immune defense against Respiratory syncytial virus (RSV). To investigate the hypothesized therapeutic role of the antioxidant GSH in the setting of RSV, we evaluated whether LGSH administered to the pups by gavage would improve pulmonary defenses against experimental RSV. Enteral LGSH administered immediately before the intranasal delivery of RSV significantly improved the capacity of AMs to phagocytose RSV in the ETOH-exposed pup (Figure 1). Furthermore, RSV growth was significantly decreased in both the BAL (Figure 2A) and the whole lung (Figure 2B) of the ETOH-exposed pups gavaged with LGSH, approaching values observed in the control + RSV group.

In vivo liposomal glutathione (LGSH) treatments attenuated Respiratory syncytial virus (RSV)-induced acute lung injury. The diminished RSV growth seen in the ETOH-exposed pups gavaged with LGSH was accompanied by decreased markers of RSV-induced acute lung injury. Indeed, whole-lung wet/dry weight ratios (Figure 3A) and lung MPO levels (Figure 3B) were significantly decreased when the ETOH-exposed pup was treated with LGSH. Similarly, LGSH treatments also attenuated PMNs influx (Gr-1+ cells) into the airspace for both the RSV-treated control and the ETOH pups, suggestive of decreased acute lung injury (Figure 4).

Liposomal glutathione (LGSH) treatments also decreased the oxidant stress in pups challenged with Respiratory syncytial virus (RSV). In the ETOH + LGSH + RSV group, the plasma AOC was significantly improved when compared to the ETOH + RSV group (Figure 5A). Likewise, LGSH treatments resulted in a significant decrease in DNA oxidant damage for both the control + RSV group and the ETOH + RSV group (Figure 5B). Similar results were observed in the GSH pool for AM, as noted by improved GSH/cell for both the control + LGSH + RSV group and the ETOH + LGSH + RSV group (Figure 6).

Liposomal glutathione (LGSH) treatments improved alveolar macrophage (AM) immune functions in pups challenged with Respiratory syncytial virus (RSV). In the control + RSV + LGSH group, there was a significant decrease in AM expression of the immunosuppression markers TGFβ1 (Figure 7A) and Arg-1 (Figure 7B) when compared to AMs from the control + RSV group. In the ETOH + LGSH + RSV group, there was also a significant decrease in the TGFβ1 (Figure 7A) and Arg-1 (Figure 7B) immunosuppression markers in the AMs when compared to the ETOH + RSV group. However, these AM markers of immunosuppression remained significantly elevated in the ETOH + LGSH + RSV group when compared to the AMs from the control + RSV group and the control + LGSH + RSV group.

## 4. Discussion

Fetal alcohol exposure is well known to adversely affect the developing newborn, most notably the developing brain [39]. However, evidence continues to mount that in utero alcohol also adversely alters multiple organs in the developing fetus [40,41], including the developing lung. As we and others have described, multiple cell types within the ETOH-exposed developing lung are at risk of alcohol-induced injury and altered pulmonary immune function [42,43,44].

Using small-animal models of in utero ETOH exposure, we have previously shown that ETOH alters immune defenses in the both the premature and the term lung, diminishes the lung’s major antioxidant GSH, and alters the immune phenotype of the resident AMs [1]. Specifically, in utero ETOH exposure induces an immunosuppressed AM phenotype, characterized by increased cellular TGFβ1, an important immunosuppressant that decreases bacterial as well as viral phagocytosis by the AM [11,12,31].

In these models, the detrimental effects of in utero ETOH exposure were modulated by the availability of the antioxidant GSH. We have also demonstrated that experimental interventions to diminish oxidant stress by maintaining or replenishing GSH status in the neonatal ETOH-exposed lung results in decreased neonatal AM TGFβ1, and more importantly, improved AM immune functions after in utero ETOH exposure. Indeed, beneficial interventional strategies such as dietary S-adenosyl methionine, a GSH precursor, during maternal ETOH ingestion [45] or exogenous inhaled GSH to the neonatal pup improved neonatal defenses against experimental bacterial infection. These studies suggest a potential therapeutic role for exogenous GSH to improve pulmonary immune defenses in the vulnerable ETOH-exposed neonate.

In adult studies, enteral LGSH diminishes markers of systemic oxidant stress and improved immune function [46]. Furthermore, a randomized control trial in healthy adults demonstrated that oral LGSH increases body stores of GSH and improves immune cell function [47]. To expand on our previous findings of ETOH-induced immune dysfunction and neonatal infection risk with either bacteria or viruses [31], the current study evaluated the potential therapeutic role for enteral LGSH in the setting of an experimental neonatal viral infection with RSV. In our neonatal mouse model, in utero ETOH-exposed pups demonstrated increased pulmonary RSV infection, accentuated markers of acute lung injury, and diminished AM phagocytosis of RSV when compared to control + RSV pups. The ETOH-induced derangements in immune defenses against RSV were accompanied by decreased GSH in the AM and increased AM immunosuppression, as demonstrated by increased TGFβ1 and Arg-1 immunostaining. Excitingly, enteral LGSH administered immediately prior to experimental RSV delivery successfully improved the ETOH-exposed pup’s pulmonary defense against RSV, as evidenced by diminished RSV growth; decreased markers of acute lung injury (lung wet/dry weight ratio, whole-lung MPO, and PMN count); decreased AM expression of immunosuppression markers (TGFβ1 and Arg-1); and restored AM in vivo phagocytosis of RSV. The role of LGSH in supplying the critical antioxidant GSH and decreasing ETOH-induced oxidant stress was demonstrated by normalization of the systemic antioxidant capacity, systemic DNA oxidation, and the AM pool of GSH.

The current study is important because newborns, particularly former premature newborns, are at risk of significant morbidity due to viral infections such as RSV [20,21]. Although we have demonstrated that in utero alcohol exposure is reported in a third of premature babies [2], the risk and severity of subsequent RSV infection in alcohol-exposed newborns remains poorly described. While RSV infections have been reported to be increased clinically in alcohol-exposed male babies [48], the mechanisms underlying the increased risk of RSV infection in the alcohol-exposed newborn requires further study. Furthermore, the potential for LGSH to augment immune defenses in the RSV-infected infant requires additional investigation.

The immune defense of airway epithelial cells is highly sensitive to alcohol and alcohol-induced oxidant stress [49,50], and these epithelial cells are the primary site for RSV infections. However, the first line of cellular defense against pathogens in the airspace is the AM, which plays a key role in engulfing and digesting pathogens, thereby minimizing the exposure of other airway cells to the pathogen. AMs also play a key role as a modulator of inflammation in the airspace through cytokine production and cellular interactions, thereby making AM immune functions critical for pulmonary defenses against RSV. Optimal AM defenses dictate effective viral clearance and significantly contribute to the resolution of pulmonary disease [23,24,51,52,53,54].

For RSV, a central role for reactive oxygen species and subsequent oxidant stress are demonstrated by significant alterations in oxidant response pathways and increases in markers of oxidative damage [55,56]. In addition to downregulation of the cellular antioxidant enzyme systems, RSV augments its own replication [57]. In contrast, antioxidant treatments attenuate both RSV-induced oxidant stress and that caused by the viral burden [25,26]. This suggests that modulation of oxidant stress represents a potential novel pharmacological approach to ameliorate RSV-induced acute lung inflammation and injury [57].

Availability of the endogenous antioxidant tripeptide GSH (glutamine, cysteine, and glycine) is critical for maintaining redox homeostasis and preventing the immune cell damage caused by reactive oxygen species. One strategy for GSH supplementation during oxidant stress is oral LGSH, which has been shown to decrease oxidant stress in HIV subjects and patients with type 2 diabetes [32,33,34]. In a mouse model of an active *Mycobacterium tuberculosis* infection, GSH depletion exacerbated the pathogen burden, but oral LGSH decreased both pulmonary oxidant stress and granuloma-promoting immune responses, resulting in decreases in the pulmonary bacterial infection [58]. Similar results were obtained in a diabetic mouse model with an active *Mycobacterium tuberculosis* infection, demonstrating that oral LGSH treatments can also improve bacterial clearance in an immunocompromised model [59]. In human clinical trials, enteral LGSH elevated antioxidant defenses and improved immune functions of natural killer cells and lymphocytes [46,47], however studies of enteral LGSH in the neonatal population are lacking.

Overall, the current study provides provocative evidence that in utero ETOH exposure results in chronic oxidant stress, which subsequently mediates dysregulation of the AM immune responses and the risk of a viral infection. Correspondingly, this increased the risk of acute lung injury associated with a viral infection. However, restoration of the GSH pool through enteral LGSH treatments improved the GSH pool in AMs and decreased the AM immunosuppression associated with in utero ETOH exposure. Similarly, the systemic oxidant stress as well as the exacerbation of the RSV infection and acute lung injury in the ETOH + RSV group were attenuated by the enteral LGSH therapy in the exposed pup. In this study, we focused on AMs and cannot rule out the possibility that other cell types also negatively affected by in utero ETOH ethanol exposure were positively impacted by the enteral LGSH therapy and contributed to the improved RSV clearance. Likewise, additional studies are needed to determine if these effects of fetal ethanol exposure on chronic oxidant stress or AM immune function improve with pulmonary maturation.

Additional studies are also needed to determine if there is a potential therapeutic role for LGSH in this vulnerable population, where timing of LGSH delivery before or at the time of the viral exposure may be critical. If these findings ultimately prove to be relevant in the clinical situation, then LGSH therapy may provide a strategy to improve AM immune responses and decrease the risk and severity of pulmonary infections in the highly vulnerable preterm infant with fetal alcohol exposure. Novel strategies such as LGSH supplements may also become particularly important in this era of antibiotic-resistant bacterial infections.

## 5. Conclusions

Fetal ethanol exposure promoted chronic oxidant stress in alveolar macrophages and systemically.Chronic oxidant stress resulted in immunosuppression of alveolar macrophages.In utero ethanol exposure impaired the capacity of pup alveolar macrophages to clear viruses.Fetal ethanol exposure exacerbated lung respiratory syncytial virus infection and acute lung injury.Enteral treatments of the pup with liposomal glutathione normalized alveolar macrophage immune responses, lung viral infection, and acute lung injury.

## Figures and Tables

**Figure 1 antioxidants-13-00137-f001:**
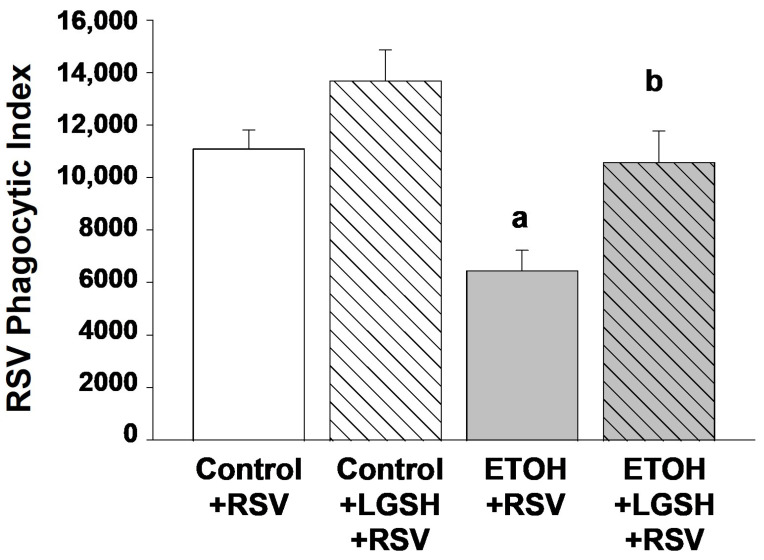
In utero ethanol (ETOH) exposure impaired in vivo alveolar macrophage (AM) phagocytosis of respiratory syncytial virus (RSV) but restored by enteral liposomal glutathione (LGSH) treatments. After breeding, the female mice were randomized to an experimental isocaloric liquid diet that contained either 25% ethanol (ETOH)-derived calories or 25% maltose–dextrin-derived calories. The diet was continued throughout pregnancy and after spontaneous term delivery. Pups were kept with their respective dams and allowed to nurse *ad libitum*. On postnatal day 10 (P10), pups from the control-fed dam and the ETOH-fed dam were then randomized to an oral gavage containing either liposomal glutathione (LGSH) (20 μL, 1.7 mg of L-glutathione) or saline (20 μL). All pups were then given intranasal injections of respiratory syncytial virus (RSV) (Nanoliter Injector; 20 µL; each nasal nare; 2 × 10^5^ PFU) before they were returned to their respective dams. After 24 h, pups received an additional dose of ±LGSH (or saline) by gavage. All pups were then euthanized for analyses after 48 h, and the alveolar macrophages (AMs) from the pups were isolated from the multiple BALs and pooled per experimental group and litter. For assessment of RSV phagocytosis by the AM in vivo, we evaluated whole-cell RSV via fluorescent immunostaining (1 h; a 1:100 dilution; Santa Cruz Biotechnology, Inc., Santa Cruz, CA, USA). Background fluorescence of unstained AMs was used to account for autofluorescence and subtracted from the RFUs obtained for the RSV phagocytosis. The RSV burden was calculated relative to the RSV fluorescence for AMs from the control + RSV group. N = 5 litters for each group. ^a^ *p* = 0.05 when compared to the control + RSV group; ^b^ *p* ≤ 0.05 when compared to the ETOH + RSV group.

**Figure 2 antioxidants-13-00137-f002:**
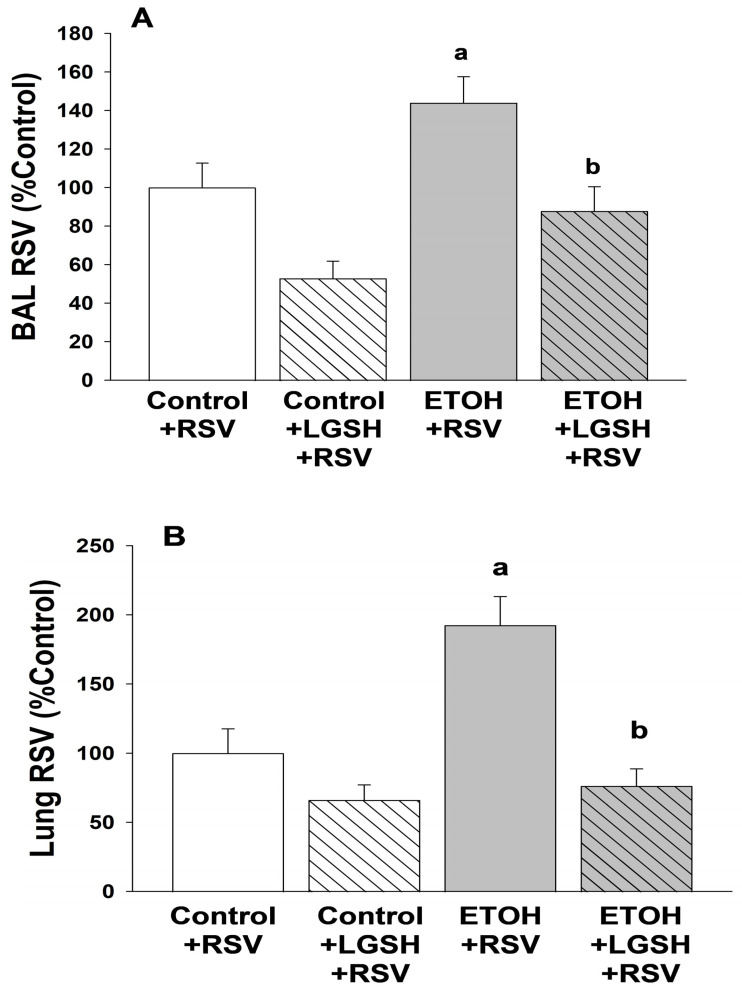
In utero ethanol (ETOH) exposure increased the respiratory syncytial virus (RSV) burden in the bronchoalveolar lavage (BAL, **A**) and lung tissue (**B**), but both were normalized by oral liposomal glutathione (LGSH) treatments. All pups were euthanized for analyses after 48 h of RSV delivery and the lungs were then serially lavaged via the catheter with 40 µL sterile saline (5 times) to remove the fluid and cells from the airspace. To determine the RSV burden in the airways and alveolar space, the multiple BALs from pups were pooled per experimental group and litter. The pooled BAL was serially diluted in phosphate-buffered saline, plated for determination of RSV growth. The RSV burden in the BAL (**A**) is presented as percentage of control plaque forming units/mL (PFU/mL). To determine the RSV burden in the lung, the frozen right upper lung lobe was weighed, sterile phosphate-buffered saline (10 times the tissue weight) was added, and then the lobe homogenized on ice. After the samples were centrifuged (2000× *g*; 10 min; 4 °C), the supernatants were serially diluted in phosphate-buffered saline and plated for determination of RSV growth. For the whole lung (**B**), RSV is presented as percentage of control of PFU/g lung tissue. N = 5 litters for each group. ^a^ *p* = 0.05 when compared to the control + RSV group; ^b^ *p* ≤ 0.05 when compared to the ETOH + RSV group.

**Figure 3 antioxidants-13-00137-f003:**
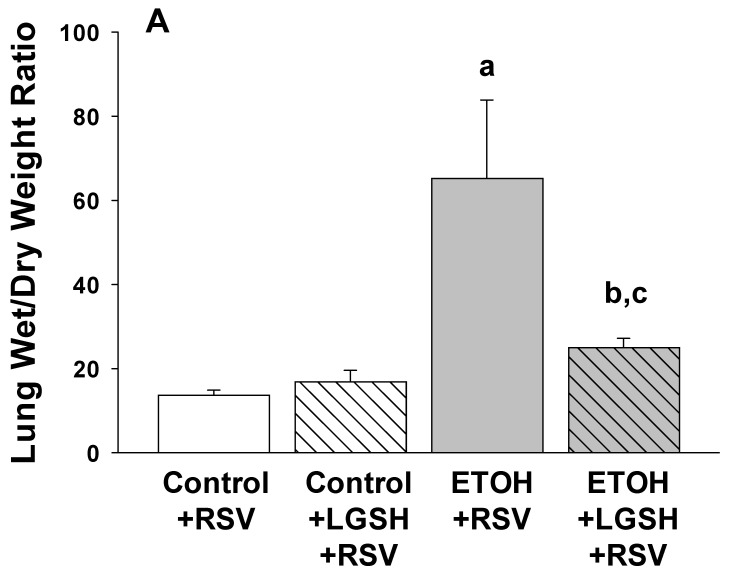
The amplified Respiratory syncytial virus (RSV) infection in the pup after in utero ethanol (ETOH) exposure resulted in increased acute lung injury, as defined by lung wet/dry weight ratio (**A**) and whole-lung myeloperoxidase (MPO) concentration (**B**), but acute lung injury was normalized by oral liposomal glutathione (LGSH) treatments. The flash-frozen right upper lobe lung samples from the neonatal pups were weighed (designated wet weight) and then reweighed after desiccation by overnight incubation at 70 °C (designated dry weight). The lung wet/dry weight ratio was determined as a marker of acute lung injury (**A**). The flash-frozen lung tissue was also evaluated for the inflammatory PMN marker myeloperoxidase (MPO) with a commercially available ELISA (product # MBS2702122, MyBioSource, Inc., San Diego, CA, USA). MPO concentrations (ng/mL) were normalized to the corresponding lung sample wet/dry weight (**B**). N = 5 litters for each group. ^a^ *p* = 0.05 when compared to the control + respiratory syncytial virus (RSV) group; ^b^ *p* ≤ 0.05 when compared to the ETOH + RSV group; ^c^ denotes *p* ≤ 0.05 when compared to the control + LGSH + RSV group.

**Figure 4 antioxidants-13-00137-f004:**
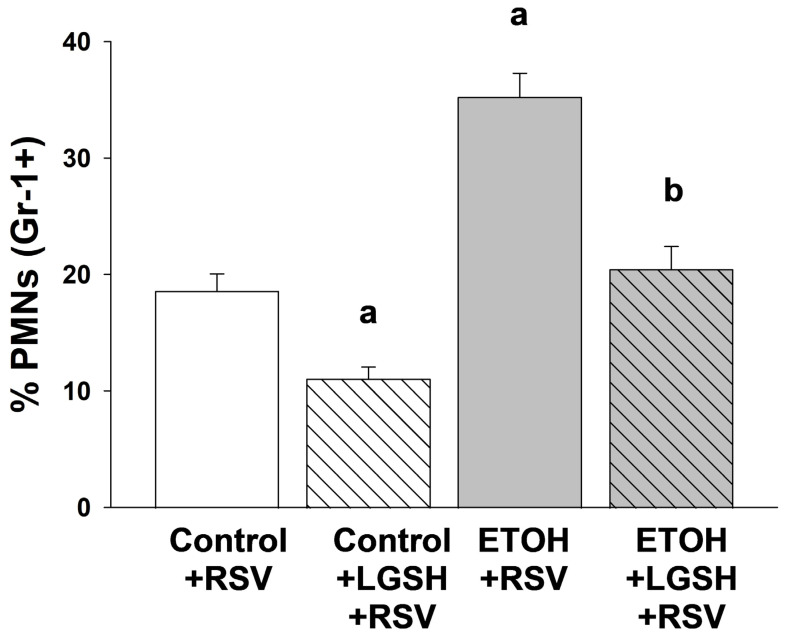
Respiratory syncytial virus (RSV) superimposed on in utero ethanol (ETOH) exposure increased polymorphonuclear leukocyte (PMN) migration into the alveolar space, but was attenuated by oral liposomal glutathione (LGSH) treatments. Immunostaining for the cell surface marker (Gr-1) was used to differentiate AM from the polymorphonuclear leukocytes (PMNs) that had migrated into the alveolar space. Cells retrieved from the lavage were plated, fixed, and incubated with the GR-1 primary antibody in a 1:100 dilution (Santa Cruz Biotechnology, Inc., Santa Cruz, CA, USA) for 1 h. After the slides were washed three times with phosphate-buffered saline over 5 min, the secondary antibody (anti-goat IgG) was added in a 1:200 dilution and further incubated for 45 min. Cellular fluorescence was quantified using fluorescence microscopy via ImagePro Plus for Windows Version 4.5 and presented as mean relative fluorescence units per cell (RFUS/cell) ± S.E.M. tallied from at least 25 cells/litter. To correct for autofluorescence, the background fluorescence of unstained AMs was subtracted from the RFUs obtained for each analysis. N = 5 litters for each group. ^a^ *p* = 0.05 when compared to the control + respiratory syncytial virus (RSV) group; ^b^ *p* ≤ 0.05 when compared to the ETOH + RSV group.

**Figure 5 antioxidants-13-00137-f005:**
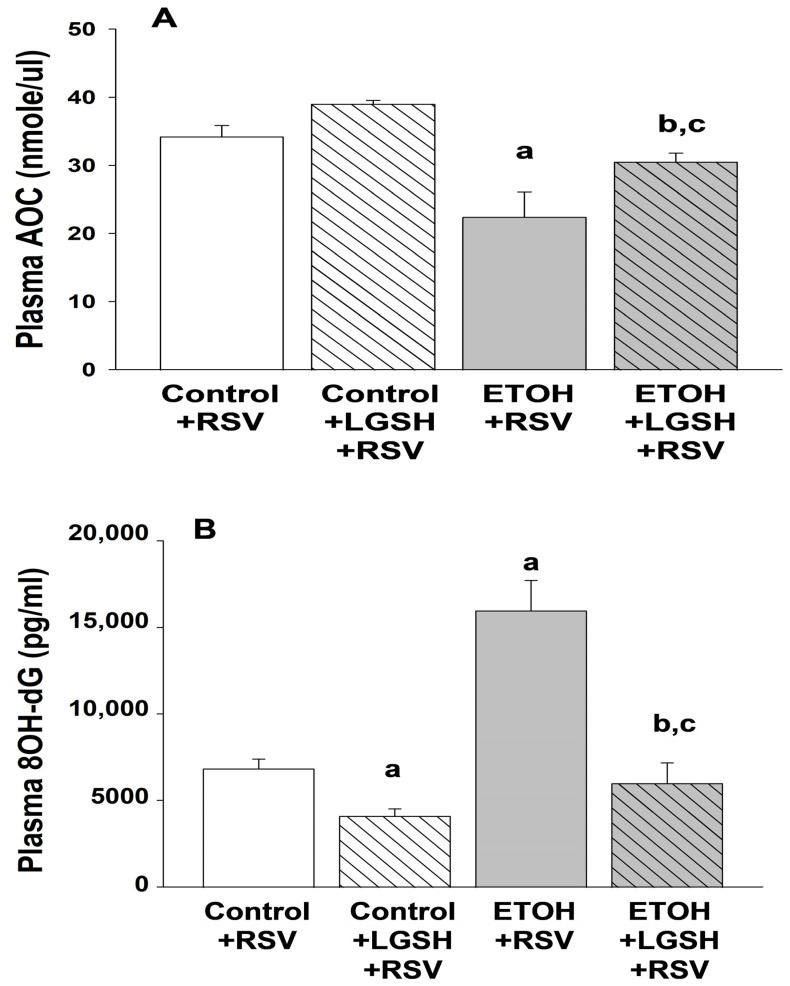
In utero ethanol (ETOH) exposure increased oxidant stress, as defined by a decrease in the plasma antioxidant capacity (AOC; **A**) and an increase in DNA oxidation (8OH-dG; **B**), but this oxidant stress was attenuated by oral liposomal glutathione (LGSH) treatments. After euthanasia with intraperitoneal sodium pentobarbital, blood samples were obtained from all pups via cardiac puncture and the samples pooled per experimental group and litter. Samples were spun and stored at −80 °C until batch analyses. Total plasma AOC (**A**) was measured via colorimetric assay (MAK187, Sigma-Aldrich, St. Louis, MO, USA) and 8-OHdG (**B**) was measured by ELISA (DNA Damage Competitive ELISA Kit, Life Technologies Corporation, Carlsbad, CA, USA). N = 5 litters for each group. ^a^ *p* = 0.05 when compared to the control + RSV group; ^b^ *p* ≤ 0.05 when compared to the ETOH + respiratory syncytial virus (RSV) group; ^c^ *p* ≤ 0.05 when compared to the control + LGSH + RSV group.

**Figure 6 antioxidants-13-00137-f006:**
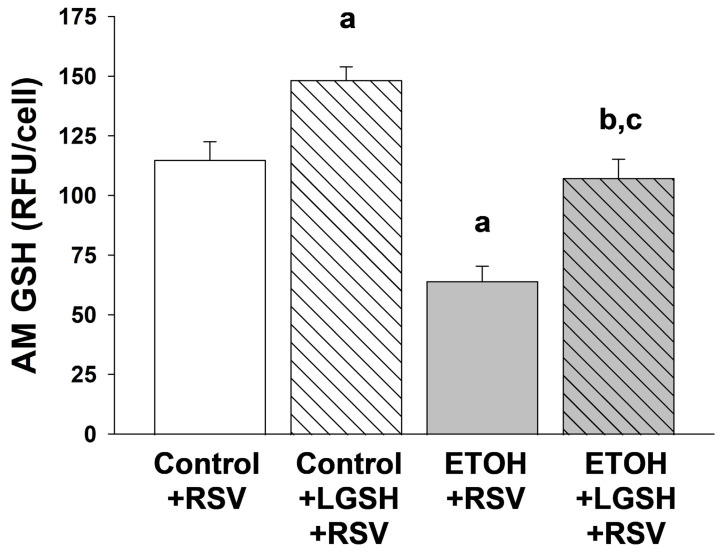
In utero ethanol (ETOH) exposure increased alveolar macrophage (AM) oxidant stress, as defined by a decrease in the glutathione (GSH) pool, but the GSH pool in the AM was restored by oral liposomal glutathione (LGSH) treatments. The freshly isolated AMs were plated and fixed with 3.7% paraformaldehyde before permeabilization with ice-cold methanol. The antioxidant GSH pool in the AMs was evaluated by whole-cell GSH via fluorescent immunostaining (1:100 dilution; Abcam, Inc^®^, Boston, MA, USA). Cellular fluorescence was quantified using fluorescence microscopy via ImagePro Plus for Windows version 4.5 [37] and is presented as mean relative fluorescence units per cell (RFUS/cell) ± S.E.M. as tallied from at least 25 cells/litter. To correct for autofluorescence, the background fluorescence of unstained AMs was subtracted from the RFUs obtained for each analysis. N = 5 litters for each group. ^a^ *p* = 0.05 when compared to the control + respiratory syncytial virus (RSV) group; ^b^ *p* ≤ 0.05 when compared to the ETOH + RSV group; ^c^ *p* ≤ 0.05 when compared to the control + LGSH + RSV group.

**Figure 7 antioxidants-13-00137-f007:**
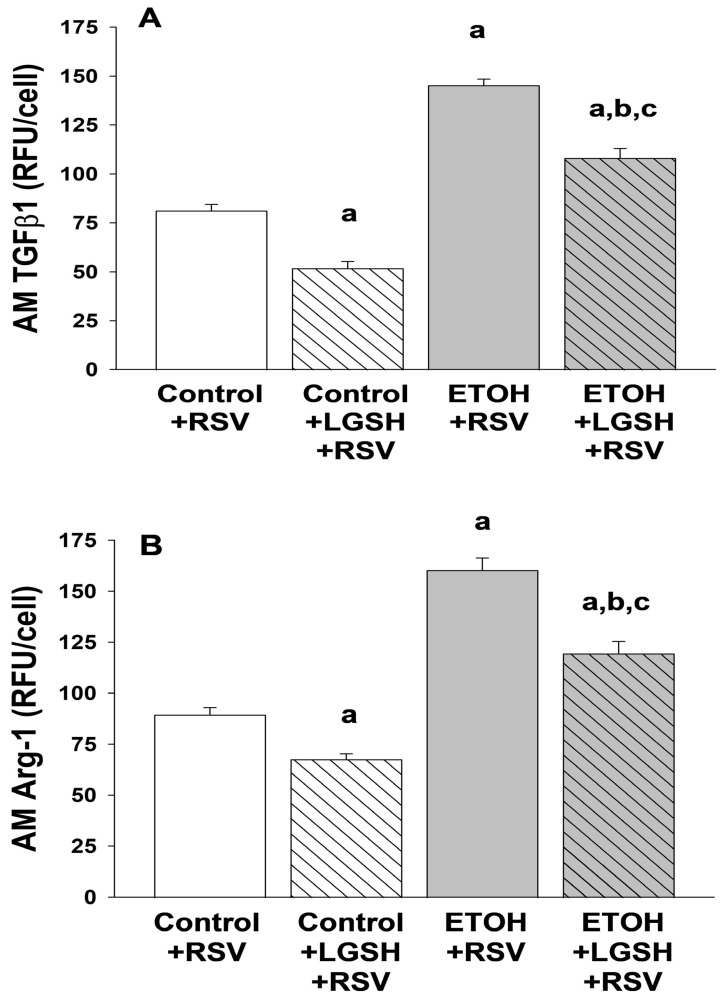
In utero ethanol (ETOH) exposure resulted in alveolar macrophage (AM) immunosuppression, as defined by an increase in transforming growth factor β1 (TGFβ1) (**A**) and arginase 1 (Arg-1) (**B**) expression, but oral liposomal glutathione (LGSH) treatments attenuated the AM immunosuppression. AM expression of TGFβ1 (**A**) and Arg-1 (**B**) were used as markers of immunosuppression. Cells were incubated with the primary antibody in a 1:100 dilution (Santa Cruz Biotechnology, Inc., Santa Cruz, CA, USA) for 1 h. After the slides were washed three times with phosphate-buffered saline over 5 min, the secondary antibody (anti-goat IgG) was added in a 1:200 dilution and further incubated for 45 min. Cellular fluorescence was quantified using fluorescence microscopy via ImagePro Plus for Windows version 4.5 [38] and is presented as mean relative fluorescence units per cell (RFUS/cell) ± S.E.M. tallied from at least 25 cells/litter. To correct for autofluorescence, the background fluorescence of unstained AMs was subtracted from the RFUs obtained for each analysis. N = 5 litters for each group. ^a^ *p* = 0.05 when compared to the control + respiratory syncytial virus (RSV) group; ^b^ *p* ≤ 0.05 when compared to the ETOH + RSV group; ^c^ *p* ≤ 0.05 when compared to the control + LGSH + RSV group.

## Data Availability

Data is contained within the article.

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
