# Peer review of "Liposomal Glutathione Augments Immune Defenses against Respiratory Syncytial Virus in Neonatal Mice Exposed in Utero to Ethanol"

_antioxidants, 2024, doi:10.3390/antiox13020137_

Round 1
Reviewer 1 Report
Comments and Suggestions for Authors
In this study, impact of oral liposomal glutathione (LSGH) treatment on ethanol-induced oxidant stress in alveolar macrophages, immunosuppression, capacity to clear viruses, Respiratory Syncytial Virus (RSV) infection and lung injury was investigated in C57BL/6 mice in utero and postnatally exposed to alcohol (0% to 12.5% and then 25% of EtOH-derived calories) through a liquid diet. Aim of the study was to “determine if a clinically relevant intervention, such as enteral LGSH could protect against an experimental pulmonary RSV infection and 2) explore the potential mechanisms by which LGSH improved AM innate immune defenses against RSV”. Mice pups (exposed and control) received LGSH (or saline) by oral gavage on day of life 10 and then intranasal injections of RSV before they were returned to their respective dams. They received an additional dose of LGSH (or saline) after 24 h, and then were sacrificed for the analyses after 48 h. The results showed that “enteral treatments of the pup with liposomal glutathione normalized alveolar macrophage immune responses, lung viral infection, and acute lung injury.
The significance of the study refers to the potential of LGSH to augment immune defenses in the RSV infected infants exposed to alcohol in utero. However, for the clinical relevance of the study it is important to investigate the timing of LGSH therapy, before or at the time of the viral exposure. In this study, LGSH delivery was performed at the time and 24 h after the viral exposure. Can the authors argue, explain such a design of the study?
In addition, with regard to real life scenario, many women stop drinking alcohol by the second trimester when they found out they were pregnant. Is it known how alcohol exposure (drinking alcohol) before and during the first trimester of pregnancy affect baby’s health (lung and immune function, for example)?
L 108-110: Please, provide a number of the Ethical Approval.
L 124, 163, 237: Instead of the “gm”, please use “g” as symbol for mass, measurement of weight. The unit symbol for gram recognized by the International System of Units is “g” and its use is highly recommended.
L 177, 179, 185, “the secondary antibody (anti-goat IgG)", “fluorescent microscopy”, “myeloperoxidase (MPO) by a commercially available ELISA”: Please, provide more detailed information that other researchers can find it (e.g., type, name of the producer with the city and country, product number, model, etc.).
L 189 and Figure 3: Please, add units of measurement for the MPO concentration (activity?).
I suggest the authors to use abbreviations in the Figure legends after defining them first.
Figure 3: Please, add “weight ratio” at y-axis (It should probably be “Lung wet/dry weight ratio”).
Figure 3, L 257, and L 345 “lung wet/dry weight”, “lung wet/dry ratios”: It should be “lung wet/dry weight ratio(s)”.
Figure 3, L 257, “whole lung MPO”: Concentration? Activity?
Reviewer 2 Report
Comments and Suggestions for Authors
Interesting study, some proposals mut be considered in the text.
Minor points:
1. Please provide more details on the diet you used, I assume you used the Lieber-DeCarli diet, credit should be given. Pair-feeding in mice is crucial due to small amounts consumed in these small animals. How did you manage?
2. Can results be explained by the low carbohydrate content of the alcohol diet?
3. Why did you use LGSH instead of GSH, question of solubility?
4. Expand on use of LGSH in human newborns prior exposed to ethanol.
5. Oxidative stress, it is microsomal or mitochondrial stress? Does MEOS with its CYP 2E1 a role?
6. Fig 3A, left side, add: weight.
Round 2
Reviewer 1 Report
Comments and Suggestions for Authors
Most of the concerns raised by reviewers have been addressed in the revised version of the manuscript.
Author Response
Thank you
Reviewer 2 Report
Comments and Suggestions for Authors
Good revision.
Author Response
Thank you